# A Study on Particle Swarm Algorithm Based on Restart Strategy and Adaptive Dynamic Mechanism

Lisang Liu [1,2], Hui Xu [1,2,*], Bin Wang [1,2], Rongsheng Zhang [1,2] and Jionghui Chen [1]

1 School of Electronic, Electrical Engineering and Physics, Fujian University of Technology, Fuzhou 350118, China; liulisang@fjut.edu.cn (L.L.); 2201905138@smail.fjut.edu.cn (B.W.); rongsheng@smail.fjut.edu.cn (R.Z.); 2221905020@smail.fjut.edu.cn (J.C.)
2 National Demonstration Center for Experimental Electronic Information and Electrical Technology Education, Fujian University of Technology, Fuzhou 350118, China
* Correspondence: xuhui@smail.fjut.edu.cn

**Abstract:** Aiming at the problems of low path success rate, easy precocious maturity, and easily falling into local extremums in the complex environment of path planning of mobile robots, this paper proposes a new particle swarm algorithm (RDS-PSO) based on restart strategy and adaptive dynamic adjustment mechanism. When the population falls into local optimal or premature convergence, the restart strategy is activated to expand the search range by re-randomly initializing the group particles. An inverted S-type decreasing inertia weight and adaptive dynamic adjustment learning factor are proposed to balance the ability of local search and global search. Finally, the new RDS-PSO algorithm is combined with cubic spline interpolation to apply to the path planning and smoothing processing of mobile robots, and the coding mode based on the path node as a particle individual is constructed, and the penalty function is selected as the fitness function to solve the shortest collision-free path. The comparative results of simulation experiments show that the RDS-PSO algorithm proposed in this paper solves the problem of falling into local extremums and precocious puberty, significantly improves the optimization, speed, and effectiveness of the path, and the simulation experiments in different environments also show that the algorithm has good robustness and generalization.

**Keywords:** restart strategy; adaptive adjustment; particle swarm optimization; spline interpolation

## 1. Introduction

With the development of robot technology, the environment is becoming more and more complex, and people's performance requirements for robots are also getting higher and higher. In a complex environment to complete the task autonomously, navigation technology is more important, and path planning is an important part of navigation technology; a good planning algorithm not only can plan the shortest path, but the cost of time, robot mechanical loss costs, maintenance costs, etc. also need to be reduced to a minimum [1]. The formatter will need to create these components, incorporating the applicable criteria that follow.

Researchers have been studying the path planning problem for many years, and have been constantly exploring and improving, with some good results. For example, the A* algorithm [2], Dijkstra [3] algorithm, RRT [4], etc. can achieve some good results in simple environments, but with the increase in environmental complexity and requirements, there will be problems such as larger computation and more memory occupation. With the emergence of intelligent optimization algorithms, more and more researchers apply intelligent optimization algorithms and their improved algorithms to path planning problems. Liu Jingsen et al. [5] proposed a bat algorithm with reverse learning and tangent random exploration mechanism, combined with cubic spline interpolation to define a smooth path based on node coding. Sun Huihui et al. [6] started from the three types of reinforcement learning motion planning methods based on value, strategy, and actor-critic, and deeply

analyzed the characteristics and practical application scenarios of deep reinforcement learning planning methods, and experimentally proved that although intelligent optimization algorithms such as gray wolf algorithm [7], ant colony algorithm [8], particle swarm algorithm, and genetic algorithm [9] can initially solve the path planning problem, these algorithms have their own shortcomings. The accuracy of the search cannot be guaranteed, and it is easy to fall into the problem of local optimization.

The particle swarm algorithm, proposed by Kennedy and Eberhart in 1995 [10], is widely used to solve various engineering problems because of its fast convergence speed, ease of implementation, and few parameters for simple modeling [11–14]. However, it also has defects such as precocious puberty, low precision, and easily falling into local optimization. Thus, many improved algorithms have been proposed in recent years. In Kang Yuxiang et al. [15], in view of the problems of precocious particle swarm algorithm and low optimization accuracy, the speed update model was improved, the adaptive particle position update coefficient was increased, and a greedy strategy was added to the algorithm process. In Panda et al. [16], in view of the rapid loss of particle swarm diversity and the problem of premature convergence, they proposed that the hybrid crossover algorithm be combined with the particle swarm algorithm to enhance the ability to explore particles and surrounding space. Ouyang Haibin et al. [17] proposed a hierarchical path planning method based on the mixed genetic particle swarm optimization algorithm, which first used the genetic algorithm improved by the artificial potential field method for primary path planning, and then used the particle swarm algorithm to optimize the path for secondary optimization. However, the method does not do a good job of fusing the two algorithms. Song et al. [18] proposed a new path smoothing method. An adaptive fractional-order velocity is introduced to enforce some disturbances on the particle. A new strategy is developed to plan the smooth path for mobile robots through an improved PSO algorithm in combination with the continuous high-degree Bezier curve. Miao et al. [19] proposed a new particle swarm optimization method. The algorithm merges two strategies, the static exploitation (SE, a velocity updating strategy considering inertia-free velocity) and the direction search (DS) of Rosenbrock method, into the original PSO.

In this paper, a particle swarm optimization algorithm (PSO) based on parameter and restart strategy improvement is proposed, and it is applied to the path planning problem. We named the proposed algorithm RDS-PSO, where R represents restart strategy, D represents dynamic adjustment, and S is for spline interpolation. The uniform distribution, inverted S-type inertia weight coefficient, cubic spline interpolation function, and enhanced control learning factor are introduced in the PSO algorithm, and a restart strategy is added to enhance the global optimization performance of the algorithm. Finally, its effectiveness was verified in an experimental environment with obstacles. Experimental results show that, compared with other path planning algorithms, the proposed RDS-PSO can achieve better results in both complex and simple environments.

## 2. RDS-PSO Algorithm

### 2.1. Standard Particle Swarm Algorithm

The PSO algorithm is a population-based optimization problem heuristic strategy proposed by Kennedy and Eberhard in 1995. The core of the PSO algorithm is to share information through individuals in the group, so that the motion of the entire group is transformed from disorder to order in the solution space problem, so as to obtain the optimal solution of the problem. The result of each optimization problem is performed by Equations (1) and (2). The first term of the velocity update Formula (1) is the inertia part, which indicates that the next move of the particle is influenced by the size and direction of the velocity of the last flight, and the inertia weight $w$ determines how much information is inherited from the previous generation, thus balancing the global and local search; the second term indicates that the subsequent move of the particle is influenced by the particle's own historical experience, and the closer the particle is to its own historical best position, the smaller the difference between the second term and the smaller the velocity. From

Formula (2), it can be seen that the next step position distance is also smaller, which at this time is conducive to local search; the third term indicates that the next action of the particle is influenced by the best particle in the group, the same as the second part, the farther the particle is from the best position in the group, the larger the difference; at this time the speed is larger, the step length in Formula (2) is also larger, which is conducive to global search. Therefore, the next step of the particle is determined by three parts: the inertial part, its own historical experience, and the group historical experience.

Particle velocity update formula:

$$V_{id}^{t+1} = wV_{id}^t + c_1 r_1 (Pbest_{id}^t - x_{id}^t) + c_2 r_2 (Gbest_{id}^t - x_{id}^t) \tag{1}$$

Position update formula:

$$x_{id}^{t+1} = x_{id}^t + V_{id}^{t+1} \tag{2}$$

where $V_{id}^t$ is the speed at which the $i$th particle flies; $t$ is the number of iterations; $d$ denotes dimensionality; $c_1$ and $c_2$ are the learning factor; $r_1$ and $r_2$ are random numbers within [0, 1] to enhance randomness; $Pbest_{id}^t$ indicates the best position of particle $i$ in the $t$ iteration; $Gbest_{id}^t$ represents the best position of the particle population in the $t$ iteration; and $w$ is the inertia weight coefficient that adjusts the search space searchability.

## 2.2. Improved Particle Swarm Algorithm

Inertia Weights

Adaptive tuning parameters have always been the focus of research on PSO algorithms. The change of inertia weight $w$ affects the position of particles, the larger the value of $w$, the stronger the global search ability, the weaker the local search ability. Several studies show that the dynamic adjustment of $w$ can improve the convergence and search accuracy of PSO. The value of $w$ can vary linearly during a PSO search [20] or dynamically as an adaptability function based on PSO performance [21]. Since the fixed and simple linear decrement strategy is not conducive to the global search of particles, this paper proposes an adaptive and dynamic weight adjustment method, that is, the inertial weight based on the sin function is introduced in the linear decrement strategy, which makes $w$ take a larger value in the early iteration period, which strengthens the algorithm's global search capability; at the same time, it takes a smaller value in the later stage, and strengthens the algorithm's local search capability.

The improved inertia weight formula is:

$$w = w_{\max} - (w_{\max} - w_{\min}) \sin\left(\frac{\pi * t}{2Itmax}\right)^2 \tag{3}$$

where $w_{\max}$ is the maximum inertia weight, $w_{\min}$ is the minimum inertia weight, $Itmax$ is the maximum number of iterations, and $t$ is the current number of iterations.

As can be seen from the above Figure 1, this improved strategy makes the inertia weights show an inverted S-shaped decreasing trend throughout the iterative process, keeping larger values in the early part of the process for a longer time, decreasing faster in the middle, and keeping smaller values in the later part of the process for a longer time. This can balance the global search and local search well.

## 2.3. Learning Factors

As important parameters in PSO, learning factors $c_1$ and $c_2$ have the effect of regulating the performance of the algorithm, which determines the influence of the particle's own historical experience and group experience on the particle motion trajectory, reflecting the information exchange between particles. $c_1$ and $c_2$ are too large or too small to facilitate particle search [22]. This paper adopts the power function to perform symmetric treatment of $c_1$ and $c_2$. The specific formula is as follows:

$$c_1 = \alpha e^w \tag{4}$$

$$c_1 = \beta e^{-w} \tag{5}$$

In order to achieve the symmetry effect, after several experiments, the two coefficients in the equation are taken as $\alpha = 0.83$ and $\beta = 2$. In the improved learning factor Formulas (4) and (5), it can be found that $c_1$ is decreasing while $c_2$ is increasing. The early focus on individual information exploration is a feasible solution. The later stage focuses on the rapid convergence of global information, which not only makes PSO have good learning ability in the optimization process, but also turns the inertia weight and learning factor into a variable, which is convenient for practical application and also strengthens the uniformity in the process of algorithm optimization.

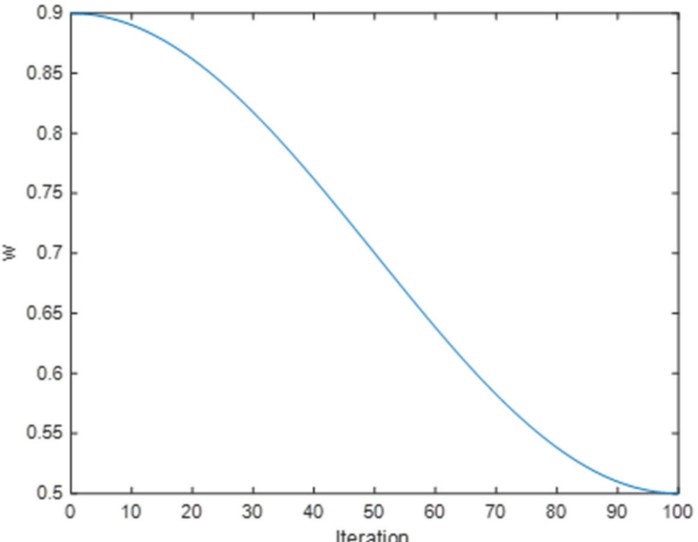

**Figure 1.** Inertia weight curve graph.

*2.4. Cubic Spline Interpolation*

In the simulation experiment, it was found that the path of the classical PSO program has many turning points, the path is not smooth enough, and the dynamic characteristics are poor during sharp turns. Thus, it is necessary to further improve the algorithm to make the algorithm more in line with the dynamic adaptability requirements of the robot.

Cubic spline interpolation is a piecewise interpolation method that can be fitted by multiple interpolation intervals based on cubic polynomials to form a smooth curve, and the robot movement path fitted with the cubic spline interpolation method is smoother.

The definition and algorithm of cubic spline interpolation are as follows:

In the interval [a, b], there are n + 1 data nodes $(x_1, y_1), (x_2, y_2), \ldots (x_n, y_n)$ that are called cubic spline functions if the following conditions are met.

Each interval $(x_i, x_{i+1})$, where i = 0, 1, $\ldots$ , n, satisfies the second cubic polynomial:

$$f_i(x) = a_i + b_i(x - x_i) + c_i(x - x_i)^2 + d_i(x - x_i)^3 \tag{6}$$

The function and its first and second derivatives are continuous at the interpolation point.

$f(x)$ commonly uses endpoint conditions that can satisfy the following three requirements:

- Free boundary: the second derivative at the endpoint is zero.
- Fixed limitation: the range value of the differential function from the beginning to the end is specified.
- Non-node boundary: the third derivative at the 2nd to the last node is continuous.

The Algorithm 1 process is:

---

**Algorithm 1** Triple spline interpolation

---

1:　　For each of these intervals it is necessary to satisfy:
2:　　$S_i(x) = a_i + b_i(x - x_i) + c_i(x - x_i)^2 + d_i(x - x_i)^3$
3:　　$S_i'(x) = b_i + 2c_i(x - x_i) + 3d_i(x - x_i)^2$
4:　　$S_i''(x) = 2c_i + 6d_i(x - x_i)$
5:　　Input parameters x, y Interpolation point n.
6:　　Calculate step size: $h_i = x_{i+1} - x_i$
7:　　for i = 1: n − 1
8:　　　Substituting the parameters into the above matrix equation
9:　　　A system of linear equations with m as the unknown is obtained
10:　　　Solve the matrix equation to find the quadratic differential value $m_i$
11:　　　Find a, b, c, d.
12:　　　In the interval $(x_i, x_{i+1})$, the Equation (6) is obtained.
13:　　　end

---

### 2.5. Particle Coding

The junction of each segment is termed a path node, and the spline curve of each segment is distinct. Cubic spline interpolation is a segmental interpolation method. The cubic spline curve is first-order continuous in nature and second-order continuous at the node; the number of path nodes denotes the maximum number of turns in the entire path; in the most challenging instance, obstacles can be avoided after 3 to 4 turns. As a result, the particle encoding in this paper is based on path nodes.

Assuming that there are path nodes $(x_{m1}, x_{m1})$, $(x_{m2}, x_{m2})$, ..., $(x_{mm}, x_{mm})$, the co-ordinates of the start point and end point are $(x_s, x_s)$, $(x_t, x_t)$, and *n* interpolation points are obtained on the interval $(x_s, x_{m1}, x_{m2}, ..., x_t)$ and $(y_s, y_{m1}, y_{m2}, ..., y_t)$ by cubic spline interpolation, and the coordinates of the interpolation points are $(x_1, x_1)$, $(x_2, x_2)$, ..., $(x_m, x_m)$. Finally, the line consisting of the path nodes, interpolation points, and the start and end points are the robot motion path we require.

### 2.6. Evaluation Function

In the path planning problem, two conditions are generally satisfied to determine whether a path is optimal or not: (i) it cannot collide with an obstacle; (ii) the path is required as short as possible.

The fitness function *F* constructed in this article is shown in Equation (7), where *L* represents the planned path length, and its mathematical expression is Equation (8), where $(x_i, x_i)$ is the coordinate of the *i* interpolation point, and *a* is a weight coefficient set to 100, which is used to exclude illegal paths. *P* is a barrier avoidance constraint function that is used to determine the safety distance; the calculation formula is shown in (9), where $R_m$ is the radius of the m-th obstacle, *m* is the number of obstacles, and *c*, *d* is the obstacle's center coordinate; the smaller the value of *P*, the higher the final path's safety factor.

$$F = L \times (1 + a \times P) \tag{7}$$

$$L = \sum_{i+1}^{n} \sqrt{(x_{(i+1)} - x_i)^2 + (y_{(i+1)} - y_i)^2} \tag{8}$$

$$P = \sum_{m=1}^{m} \left( MAX \left( 1 - \frac{\sqrt{((x_i - c)^2 - (y_i - d)^2)}}{R_m}, 0 \right) \right) \tag{9}$$

### 2.7. Restart Strategy

A restart strategy is introduced under the above improvement circumstances in order to increase the algorithm's optimization abilities and overcome the problems of local optimization and precocious puberty. Huberman et al. were the first to use the restart technique to a stochastic optimization algorithm in 1997 [23]. It has become a standard

strategy in stochastic optimization algorithms, and it is frequently used to boost algorithm performance [24]. By reinitializing the generation of fresh potential particles, you can avoid getting into a local ideal scenario.

In this paper, an iteration threshold is set in the process of the algorithm. If the optimal solution is not improved in the process of successive H-generation iterations, the optimal solution will be retained at this time and reinitialized into the next iteration. The improvement strategy enables the algorithm to effectively jump out of the local optimum, enhance the global search capability of the algorithm, and avoid premature maturity of the algorithm.

### 2.8. RDS-PSO Algorithm

Through the above comprehensive improvements, the inverted S-type inertia weights better balance the global and local search ability of the algorithm, and the dynamic learning factor not only strengthens the learning ability of the algorithm in the optimization process, but also combines the inertia weights and the learning factor into one variable, which is convenient for practical applications. On this basis, the cubic spline interpolation method is introduced to smooth the path, which improves the defect of the unsmooth path and enables the robot to better adapt to the real environment. For the problem that PSO is prone to falling into local optimum and premature maturity, a restart strategy is introduced by combining the above improved parameters, and the improved strategy enhances the algorithm's optimization-seeking ability and improves the problems of premature maturity and falling into local optimum. We call the proposed algorithm RDS-PSO.

The basic steps of the RDS-PSO algorithm are as follows.

Step 1: The number of path nodes and the number of interpolation points are determined according to the specific environment, and the starting and ending points are determined.

Step 2: Set the parameters, initialize the population and particle velocity, and initialize the population distribution.

Step 3: The coordinates of the interpolation points in the x and y directions are calculated for each particle using the cubic spline interpolation method.

Step 4: Calculate the adaptation value using Equation (7)

Step 5: The parameters are updated according to Equations (1)–(5), respectively, and update the local optimal value $Pbest_{id}^{t}$ and the global optimal value $Gbest_{id}^{t}$ and save it.

Step 6: According to Equation (9), we confirm whether the updated particle intersects with the obstacle, and apply algorithm 1 to obtain a path consisting of path nodes, interpolation points, and start-end connections after the update.

Step 7: In the iteration process, determine whether the restart condition is met. If the restart condition is met, the optimal path is kept at this time, reinitialized, and steps 1 to 6 are executed again; if not, the number of iterations is increased by 1 until the maximum number of restarts is reached, the algorithm ends, and the path is output.

The specific flowchart is shown in Figure 2.

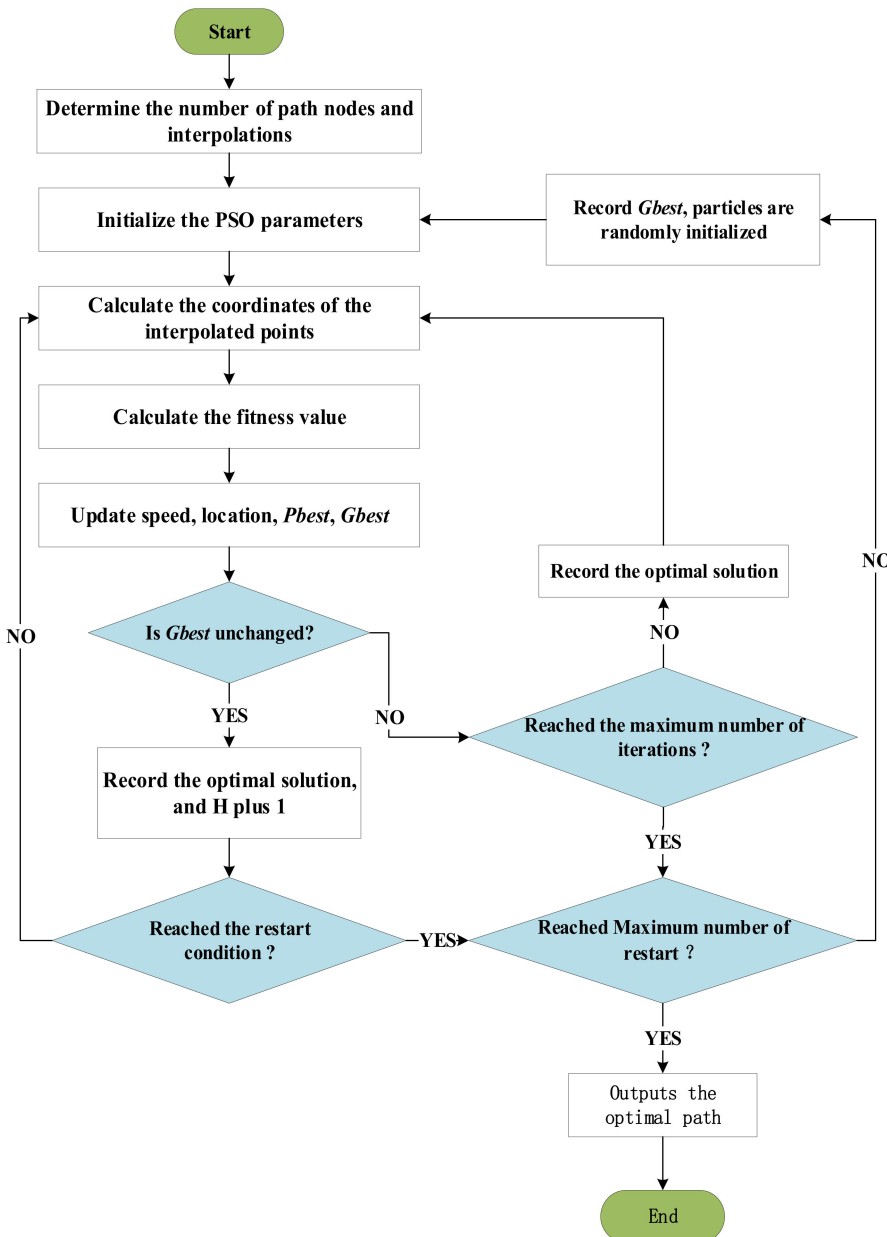

**Figure 2.** Flowchart of the RDS-PSO algorithm.

## 3. Experiments and Analysis of Results

### 3.1. Experimental Environment and Parameter Settings

The RDS-PSO algorithm and the standard particle swarm algorithm (PSO), the Improved PSO (RandWPSO-SP) based on random inertia weights and cubic spline interpolation [25], and the improved particle swarm optimization algorithm (IPSO) proposed in the literature [26] were experimentally compared and analyzed to verify the effectiveness and advancedness of the proposed algorithm in solving the robot path programming problem. This evaluates the algorithm's performance in terms of path planning for robots.

In order to ensure the objectivity and fairness of the experiment, all algorithms use the same software and hardware platform for experimentation, the simulation environment is Windows 10, Core i5, CPU (2.4 GHz), memory 12 GB, programming environment MATLAB R2019b. In order to ensure the authenticity of experimental data, 30 independent experiments on each algorithm, the experimental data were averaged.

In the simulation experiment, the parameters of the four algorithms, such as population size and maximum number of iterations, were consistent with $Itmax = 100,$

$Npop = 150$, In the standard PSO, the inertia weights and learning factors, $w = 0.9$, $c_1 = 1.5$, $c_2 = 1.5$, RandPSO-SP and the same parameter settings in this algorithm are consistent, $w_{max} = 0.9$, $w_{min} = 0.4$, the number of cubic spline interpolation points is set to 100, and the boundary is non-node boundary. Among them, the learning factor regulation parameters in the algorithm of this paper are $\alpha = 2$, $\beta = 0.83$.

In order to verify the universality of the algorithm in the path planning problem, the simulation experiment is carried out on MATLAB.

### 3.2. Experiments in Map 1

There are many obstacles in map 1, where obstacles are represented by blue circles. As can be seen from Figure 3, compared with the other three algorithms, the RDS-PSO of this algorithm has a shorter path, the least inflection point, and because the obstacles are more scattered, the best path is almost straight, and the other paths are smoother, which is due to the use of cubic spline interpolation, so the path is smoother.

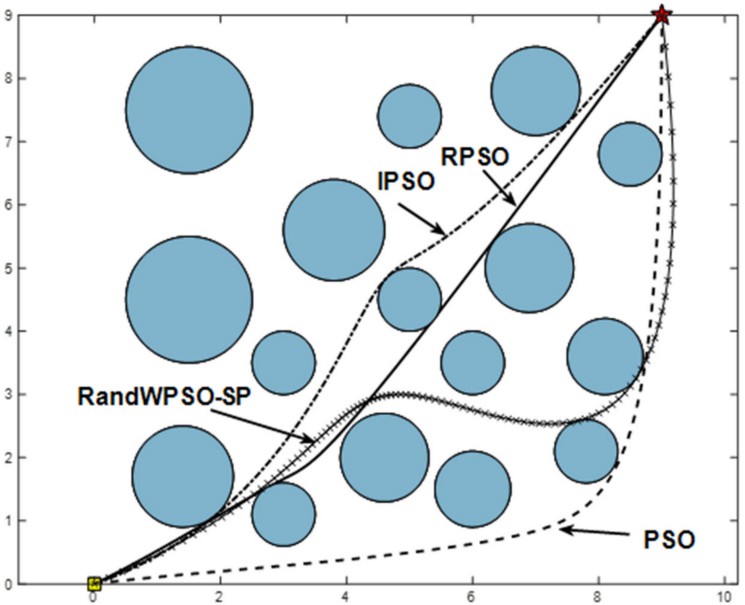

**Figure 3.** Comparison of path planning.

The iterative process of RDS-PSO is shown in Figure 4, and it can be seen that the algorithm has performed two restarts and finally found the optimal path because the algorithm has added a restart strategy. When the algorithm stagnates, it can be considered that the algorithm falls into local optimization; at this time, a new randomly distributed particle is added, combined with the inverted S-type inertia weight and the learning factor improvement method to improve the algorithm search ability, and also uses the characteristics of PSO convergence speed to shorten the iteration time; and restart multiple times to find the optimal path to achieve the purpose of jumping out of the local optimal.

The fastest convergence of IPSO in iterative Figure 5 is due to the addition of enhanced learning factors, but it can be seen in Table 1 that the algorithm is less robust and difficult to jump out when it falls into local optimality. RandWPSO-SP and PSO converge at the same rate, converging around 10 generations, but the optimal path was not found. RandWPSO-SP is too random; although the perturbation is obvious, it is easy to miss the optimal solution, and when the particles converge, it is not easy to jump out of the local optimal.

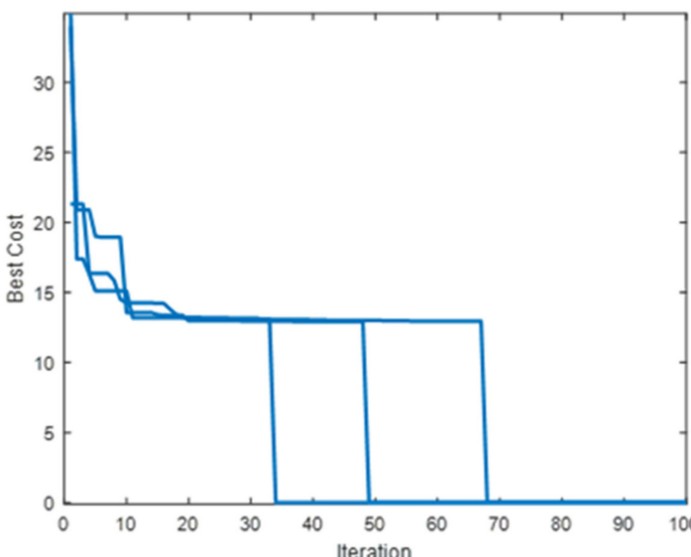

**Figure 4.** RDS-PSO iteration.

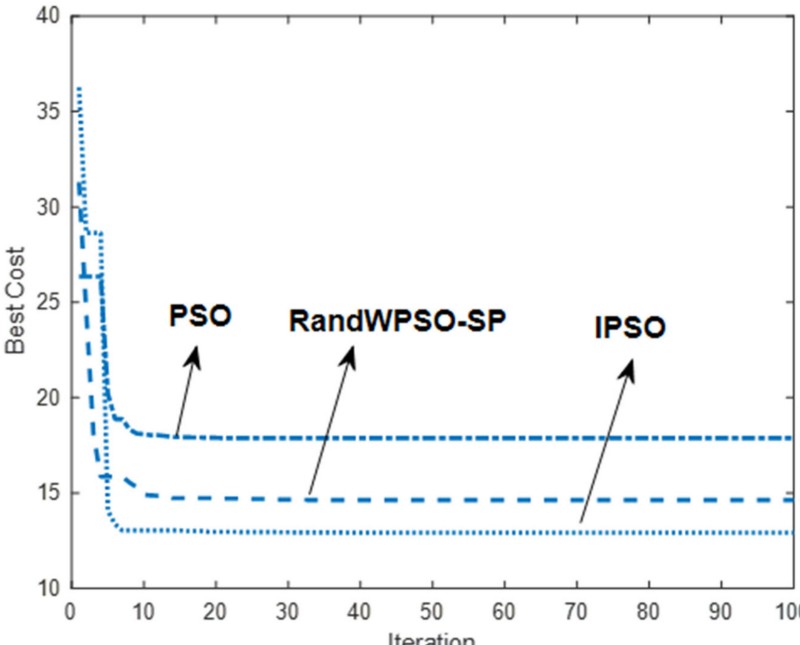

**Figure 5.** Iteration of three algorithms.

The data in Table 1 are the path results obtained by running the algorithm independently for 30 times in map 1, and an accuracy rate is introduced in the table as an evaluation index to judge the stability of the algorithm, that is, to find the optimal solution or the suboptimal solution is to find the correct path. It can be seen from the table that the average path length, the worst path, and the average simulation run time of the RDS-PSO are better than the other three algorithms, and the four algorithms have found the optimal path, but the RDS-PSO has the highest accuracy rate, only once did not reach the optimal value, which is due to the introduction of the restart strategy. When it falls into the local optimal, you can find a new solution in time, combined with the improved inverted S-type inertia weight and symmetric learning factor to enhance the search ability while improving the convergence speed. In this way, many optimizations are sought in a short period of time, which greatly enhances the optimization ability of the algorithm, and the optimization results are more stable.

**Table 1.** Comparison of algorithm performance.

| Algorithm | Longest Path | Shortest Path | Average Path | Average Time (s) | Accuracy |
|---|---|---|---|---|---|
| PSO | 12.89 | 15.34 | 13.6 | 29.86 | 47% |
| IPSO | 12.9 | 15.85 | 13.96 | 30.7 | 57% |
| RandWPSO-SP | 12.89 | 15.45 | 13.47 | 29.46 | 60% |
| RDS-PSO | 12.89 | 13.16 | 12.94 | 29.10 | 94% |

### 3.3. Experiments in Map 2

In the experimental map 2, the environment is more complex. With continuous obstacles, there is less room at the beginning, fewer paths to choose from, and it is easier to fall into local extremums; therefore, the ideal path must span a tighter area.

As can be seen in Figures 6 and 7, the path prepared after two RDS-PSO restarts is the shortest and smoothest. As can be seen in Figure 8, RandWPSO-SP has multiple jumps out of the native extremum, which is due to the addition of random inertia weights, which strengthens particle randomness. While the ultimate designing path is also shorter, the shortest path is not found, indicating that the algorithmic rule is ineffective in improving performance. Around the twentieth generation, IPSO and PSO merged. IPSO discovered a more robust path, owing to the employment of linear decreasing inertia weights and unified learning factors to improve algorithmic rule search performance. However, the convergence speed is swift, and the algorithmic rule search performance is improved.

Table 2 shows the path results of the four algorithms running independently 30 times in map 2. It can be seen from the table that the optimal solutions of the four algorithms are the same, but the worst solutions are very different, reflecting the difference in the optimization ability of the algorithms. Compared with experimental map 1, experimental map 2 is more complex, so the accuracy of the four algorithms is reduced. The average time of RDS-PSO is slightly longer, which is caused by the restart mechanism, but the average path length and accuracy rate are the best of the four algorithms, indicating that the optimization performance and robustness of the algorithm have been greatly improved.

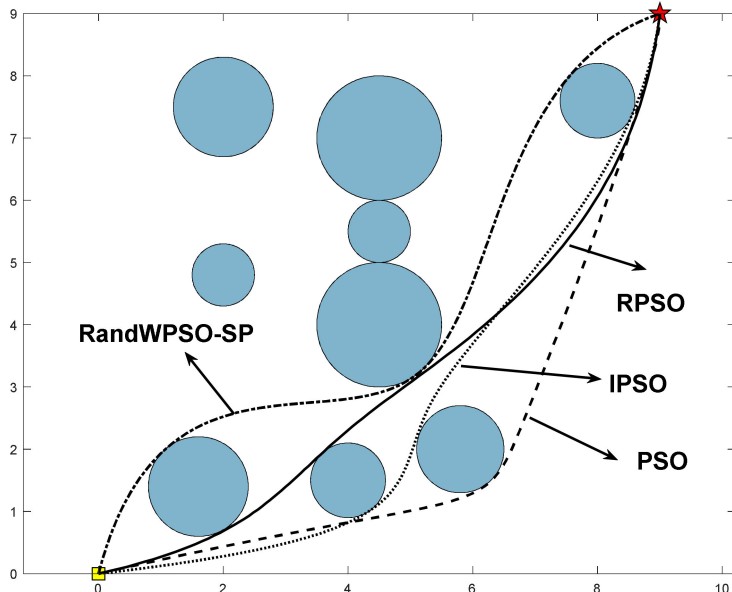

**Figure 6.** Comparison of path planning.

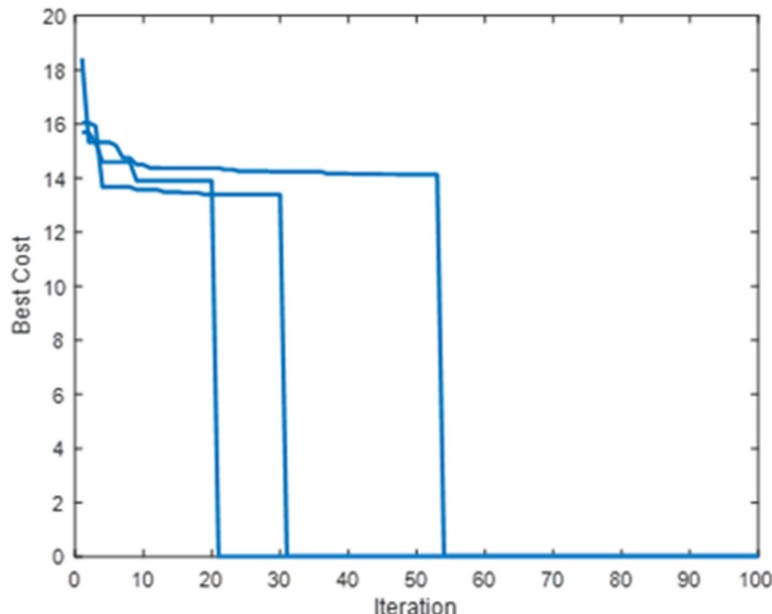

**Figure 7.** RDS-PSO iteration.

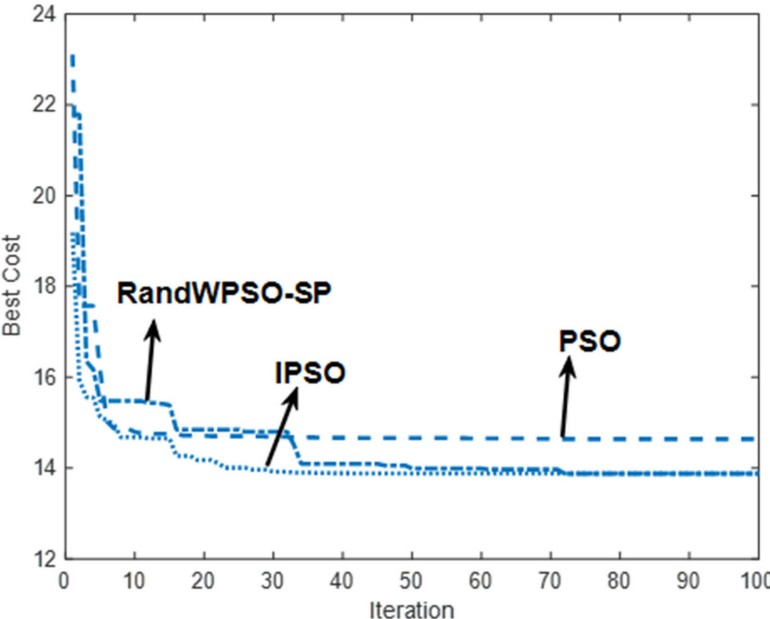

**Figure 8.** Iteration of three algorithms.

**Table 2.** Comparison of algorithm performance.

| Algorithm | Longest Path | Shortest Path | Average Path | Average Time (s) | Accuracy |
|---|---|---|---|---|---|
| PSO | 13.25 | 16.45 | 14.2 | 26.9 | 20% |
| IPSO | 13.28 | 14.58 | 14.00 | 26.2 | 40% |
| RandWPSO-SP | 13.29 | 15.24 | 14.03 | 25.76 | 50% |
| RDS-PSO | 13.25 | 14.13 | 13.58 | 29 | 80% |

### 3.4. Experiments in Map 3

Considering the diversity of actual obstacles, if all types of obstacles are expanded into circles, the feasible route may disappear, so this paper designed a third map for experimentation, as shown in Figures 9 and 10 below.

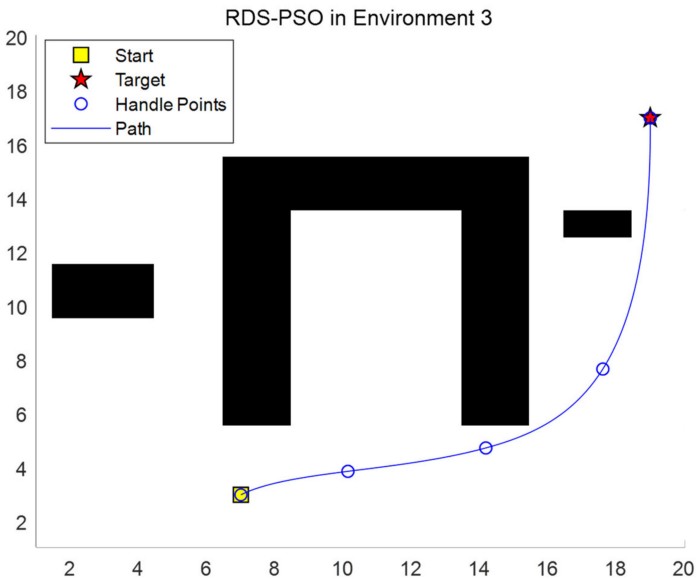

**Figure 9.** Path planning of RDS-PSO in map 3.

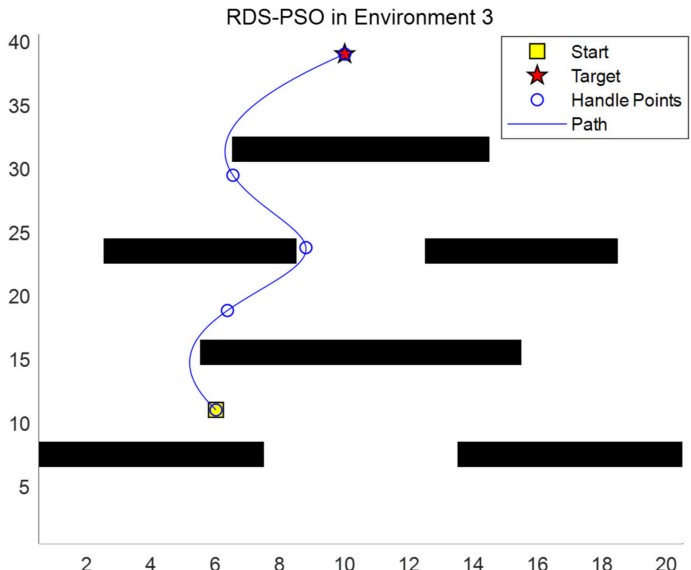

**Figure 10.** Path planning of RDS-PSO in map 3.

In experimental map 3, slender strips of various sizes are set up to evaluate the algorithm's universality and stability. Where-type barriers are used to see if the algorithm will fall into a state of local optimality. The final path fully avoids obstructions, is not "misled" by the center section of the gate, and chooses the best path, as can be seen from the planned route. The algorithm successfully avoided the obstacle, found the best path, and the path is also very smooth, as shown in Figure 10. Figure 10 enlarges the map range and sets up a continuous overlapping long bar obstacle. The starting point to the end point requires multiple turns, increasing the difficulty of planning. This report also confirms the search performance, ubiquity, and trustworthiness.

## 4. Conclusions

In this paper, an improved particle swarm algorithm combined with cubic spline interpolation is proposed to solve the robot path planning problem. For the "precociousness" in the basic PSO and some improved algorithms, the search ability is poor, it is easy to fall into local extremums, and it is difficult to jump out, resulting in problems such as search

stagnation. First of all, the key parameters of PSO are improved, a new inverted S-type inertia weight and symmetric learning factor are introduced, and these three parameters are unified into one variable, which is convenient for practical application, improves the global optimization ability of the algorithm, and also improves the uniformity in the process of algorithm optimization, and enhances the search performance of the algorithm. At the same time, combined with the characteristics of the fast convergence speed of the particle swarm algorithm, a restart strategy is introduced, and when the algorithm search is stalled, it is reinitialized with random particles, which makes it easier for the algorithm to jump out of the local extremum, and also solves the problem of not being able to find a solution due to "precocious puberty". On this basis, the path nodes in the cubic spline interpolation are encoded as individual particles, so that the PSO and cubic spline interpolation method are combined with the robot path planning to plan a smooth path. An experimental comparison of four algorithms was carried out in two environments, and RDS-PSO was tested in complex environments, and the experimental results showed that the RDS-PSO improved algorithm in this paper had better solution performance under the same time, the shortest path of planning, the highest success rate, and the more stable algorithm, which proved the effectiveness and superiority of the improved algorithm in path planning problems.

**Author Contributions:** All of the authors contributed extensively to the work. H.X. proposed the key ideas; H.X. analyzed the key contents using a simulation and wrote the manuscript; L.L. obtained the financial support for the project leading to this publication; B.W., R.Z. and J.C. modified the manuscript. All authors have read and agreed to the published version of the manuscript.

**Funding:** This work was supported in part by the Natural Science Foundation of Fujian Province under Grant 2019J01773, in part by the Initial Scientific Research Fund of FJUT under Grant GY-Z12079, Grant GY-Z21036, and Grant GY-Z20067.

**Acknowledgments:** The authors would like to thank the anonymous reviewers for their valuable comments.

**Conflicts of Interest:** The authors declare no conflict of interest.

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
