# Peer review of "A Study on Particle Swarm Algorithm Based on Restart Strategy and Adaptive Dynamic Mechanism"

_electronics, doi:10.3390/electronics11152339_

Round 1

Reviewer 1 Report

The authors study on the particle swarm algorithm and made clear statement of comparison among PSO, IPSO, RDS-PSO, and RandWPSO-SP. It requires lots of improvement on the manuscript.

1.    In general, all figures need titles and short descriptions. The font of labels needs to increase. Three curves need labels in figure 4.

2.    In line 97, the author said that ‘At this point, … by its own historical experience’. I don’t understand what ‘own historical experience’ means. Would you please make a short description?

3.    In line 139, the formula (5) is a copy of formula (4). Please correct that.

4.    In line 140, please provide the reason to pick 0.83 and 2 for alpha and beta.

5.    The section title 3.4 and 3.5 are both ‘Particle coding’. Please correct that.

6.    In line 267 and 268, please provide the reason to pick 1.5, 0.9 and 0.4.

7.    In figure 10, the blue curve is plotted with very few points. The full path of curve could be different as shown in figure 10. Please provide more evidence.

Reviewer 2 Report

The works considers the optimal movement of robots through an area with obstacles. A particle swarm algorithm is developed to solve the path planning problem. The algorithm is based on the work by Kennedy and Eberhart (1995). It improves the original approach in various aspects. Numerical experiments document the effectiveness of the modifications. I have some suggestions for improvement:

* The work by Kennedy and Eberhart is not referenced correctly. Reference [10], which is given as a source, is not the original work, neither does it mention Kennedy and Eberhart explicitly.

* Eq. (1) and (2) deserve a more detailed explanation. I found the use of subscripts and superscripts quite confusing. For example, the symbol V_{id}^t explained in line 104 does not appear in the equations, at all. The original matrix notation by Kennedy and Eberhart is more transparent.

* Although the language of the text is good in general, there are a few paragraphs with very confusing sentences, which look a bit as if generated by an automatic translator. Examples:

 -line 114: "...the dynamic adjustment of can improve the convergence..."

 -line 115: "The value of can varies linearly ..."

 -line 119/120: "the linear decrement strategy, which makes take a larger value in the early iteration period..."

 I recommend another proofreading of the text.

* line 195: "an is a weight coefficient" should probably read "a is a weight coefficient"

Round 2

Reviewer 1 Report

The author has improved the manuscript according to the review comments. It is a good manuscript now.

Author Response

非常感谢您的评论意见!

请参阅附件
